# Health-related quality of life of the adult COVID-19 patients following one-month illness experience since diagnosis: Findings of a cross-sectional study in Bangladesh

**Md. Ziaul Islam** [1‡]*, **Baizid Khoorshid Riaz** [2‡], **Syeda Sumaiya Efa** [3‡], **Sharmin Farjana** [4‡], **Fahad Mahmood** [1‡]

1 Department of Community Medicine, National Institute of Preventive and Social Medicine (NIPSOM), Dhaka, Bangladesh, 2 Department of Public Health and Hospital Administration, National Institute of Preventive and Social Medicine (NIPSOM), Dhaka, Bangladesh, 3 Diabetic Association of Bangladesh, USAID's Alliance for Combating TB in Bangladesh, BADAS TB Initiative in Bangladesh, Dhaka, Bangladesh, 4 Department of Obstetrics and Gynecology, Shaheed Suhrawardy Medical College Hospital, Dhaka, Bangladesh

‡ MZI had the highest contriburion to this work. BKR, SSE, SF and FM contributed equally to this work.
* dr.ziaul.islam@gmail.com

**Data Availability Statement:** All relevant data are within the paper and its Supporting Information files.

## Abstract

The coronavirus disease 2019 (COVID-19) stances an incredible impact on the quality of life and denigrates the physical and mental health of the patients. This cross-sectional study aimed to assess the health-related quality of life (HRQOL) of COVID-19 patients. We conducted this study at the National Institute of Preventive and Social Medicine (NIPSOM) of Bangladesh for the period of June to November 2020. All the COVID-19 patients diagnosed by real-time reverse transcriptase-polymerase chain reaction (RT-PCR) assay in July 2020 formed the sampling frame. The study enrolled 1204 adult (aged >18 years) COVID-19 patients who completed a one-month duration of illness after being RT-PCR positive. The patients were interviewed with the CDC HRQOL-14 questionnaire to assess HRQOL. Data were collected by telephone interview on the 31st day of being diagnosed and by reviewing medical records using a semi-structured questionnaire and checklist. Around two-thirds (72.3%) of the COVID-19 patients were males and a half (50.2%) were urban residents. In 29.8% of patients, the general health condition was not good. The mean (±SD) duration of physical illness and mental illness was 9.83(±7.09) and 7.97(±8.12) days, respectively. Most of the patients (87.0%) required help with personal care, and 47.8% required assistance with routine needs. The mean duration of 'healthy days' and 'feeling very healthy' was significantly lower in patients with increasing age, symptoms, and comorbidity. The mean duration of 'usual activity limitation', 'health-related limited activity', 'feeling pain/worried', and 'not getting enough rest' were significantly higher among patients' having symptoms and comorbidity. 'Not so good' health condition was significantly higher in females (OR = 1.565, CI = 1.01–2.42) and those having a symptom (OR = 32.871, CI = 8.06–134.0) of COVID-19 and comorbidity (OR = 1.700, CI = 1.26–2.29). Mental distress was significantly higher among females (OR = 1.593, CI = 1.03–2.46) and those having a symptom (OR =

**Funding:** The author(s) received no specific funding for this work.

**Competing interests:** The authors have declared that no competing interests exist.

4.887, CI = 2.58–9.24). Special attention should be given to COVID-19 patients having symptoms and comorbidity to restore their general health, quality of life, and daily activities.

## Introduction

Coronavirus is a newly emergent severe acute respiratory syndrome coronavirus 2 (SARS-CoV-2) virus, which presented as an outbreak of pneumonia of unknown cause [1]. The virus causes coronavirus disease 2019 (COVID-19), which was declared as a global emergency on the 30th January 2020 and as a pandemic on the 11th of March 2020 by the World Health Organization (WHO) [1]. Several extreme public health measures (like home confinement, the lockdown of cities, limited human mobility, an extension of national holidays, and closure of academic institutions, etc.) and diverse medical measures (like isolation, quarantine, hospitalization, etc.) are adopted worldwide to prevent transmission and to minimize the health effects of COVID-19 patients. All these measures pose a negative impact on daily living, social participation, and life satisfaction of the patients [2]. Concerning all these public health interventions, the physical activity of COVID-19 patients decreases and impairs their psychological health [3].

Bangladesh faces multidimensional challenges with the pandemic of novel corona virus-2 (n-Cov-2). The infection rate remained low until the end of March 2020 but saw a steep rise in the month of April in the country. Up to 27th June 2022, total cases of COVID-19 were more than nineteen lacs, and total deaths were more than twenty-nine thousand [4]. However, concerns have been raised because the insufficiency of testing assays may leave many cases undetected [5]. Being a lower-middle-income and densely populated country, Bangladesh is struggling to cope with this pandemic situation. Apart from this, the country faces significant challenges in combating COVID-19 because it also houses a million homeless persons and refugees in sprawling slums and refugee camps that are conducive to aggravating the pandemic [5].

Based on a synthesis of the scientific literature and advice from its public health partners, the Centers for Disease Control and Prevention (CDC) has defined Health-related Quality of Life (HRQOL) as "an individual's or group's perceived physical and mental health over time" [6]. It is a multi-dimensional concept for examining the impact of health status on quality of life [7]. Studies also reported that poor body immunity impairs the physical and mental health of the patients [8]. The elevated psychological problems and decreased quality of life across nations and occupations reflect the forthcoming worst mental and physical health situations for the victims of infectious diseases [8]. As a severely infectious disease, COVID-19 affects the health and quality of life of the victims, who are likely to have a lower HRQOL [9].

Several studies explored the HRQOL of COVID-19 patients using different HRQOL measuring tools. A cross-sectional online survey was conducted on ethnic Chinese using the Chinese version of short form-8 (SF-8) to measure the HRQOL. And the study revealed that patients' average physical component summary score (PCS) was 75.3 (Standard deviation, SD = 16.6), and mental component summary score (MCS) was 66.6 (SD = 19.3). More than half of the patients (53.0%) reported a moderate level of stress [2]. Another study conducted in Wenzhou, china used the Chinese version of the Short-Form 36-item questionnaire (SF-36) to measure the HRQOL. The study identified that age was negatively associated with physical function but positively associated with vitality. Physical function, bodily pain, and role-emotional were negatively associated with the female sex [10]. A study in Vietnam revealed that being female, having chronic conditions, and being elderly (aged ≥60 years) were associated with lower HRQOL scores [8].

The present study has performed a quantitative examination of the of life quantitatively in terms of duration of physical health, mental health, and limitation of health-related activities by using the CDC HRQOL-14 questionnaire. The key benefit of using this questionnaire was that it has continuous, cardinal, and bounded mathematical properties, which permits the use of different statistical analysis methods. This instrument didn't use a summary score and consider psychometrically derived or preference-based weights like other health profiles. It is an instrument with defined test characteristics, which is easy to administer and publicly available. It had a significant benefit in measuring HRQOL of COVID-19 patients as it is validated for telephone interviews [11, 12]. So, data collection with this instrument through telephone interviews was very realistic during the COVID-19 pandemic when maintenance of social distance was mandatory.

To date, no comprehensive study investigated how severe the impact of COVID-19 is on the quality of life of the patients in the context of Bangladesh. The present study intended to assess the HRQOL of adult COVID-19 patients considering both physical and mental illness in Bangladesh.

## Materials and methods

### Study design and setting

This cross-sectional study aimed to assess the HRQOL of the COVID-19 patients following their one-month illness experience since diagnosis by real-time reverse transcriptase-polymerase chain reaction (RT-PCR) test for SARS-COV2. We considered the one-month duration of illness experience of COVID-19 patients after RT-PCR test positive and used the CDC-HRQOL questionnaire to assess their HRQOL. Both recovered (who became RT-PCR negative within 30 days) and non-recovered (who remained RT-PCR positive after 30 days since diagnosis) patients who had completed one month of illness after diagnosis by RT-PCR test were eligible for the current study. We conducted the study at the National Institute of Preventive and Social Medicine (NIPSOM) of Bangladesh from June to November 2020. NIPSOM is the apex public health institute in the country, holding the central laboratory designated by the Ministry of Health and Family Warfare of the Bangladesh government for COVID-19 diagnosis.

### Study population

The study population included all adult COVID-19 patients diagnosed by RT-PCR assay at the central laboratory of NIPSOM. We enrolled the patients of both sexes, aged >18 years and who completed the one-month duration of illness after being diagnosed by RT-PCR test. The study excluded those patients who had no contact number or wrong contact number (Telephone/Cell); who didn't respond to a phone call on three separate occasions at three different times on the $31^{st}$ day after being RT-PCR positive; who were unwilling to participate; and patients who had incomplete interviews. We collected data through telephone interviews, and there was no scope to investigate whether the clinical characteristics of the individuals excluded from the study were similar or different from the individuals who were included. We excluded some individuals from the study who did not respond to our telephone calls and whose telephones were inactive. As a result, we couldn't collect data on the severity and couldn't be concerned about omitting any patient with particular severity. We retrieved the COVID-19 patients using the records of the central laboratory of NIPSOM.

## Sample size and sampling

All the COVID-19 patients diagnosed by the NIPSOM laboratory in July 2020 (from 1st to 31st July) before the data collection period were selected purposively for data collection. The central laboratory of NIPSOM diagnosed a total of 1342 COVID-19 patients in July 2020. Out of them, 1204 patients were enrolled in the study following the selection criteria. The selection criteria of the study included: (i) Completion of the one-month duration of illness after being RT-PCR positive; (ii) Adults aged >18 years; and (iii) Capable of giving informed consent. So, the final sample size of the study was 1204 who were enrolled following the purposive sampling technique. According to Health Information System (HIS), Directorate General Health Services (DGHS), Bangladesh, the month of July saw a total of 75,955 confirmed COVID-19 patients, out of which 53,062 were male while 22,893 were female [13]. We used these data as a reference for assessing representativeness.

## Data collection

For data collection, we recruited 15 data enumerators and three supervisors, and all of them were physicians and had a Master of Public Health (MPH) degree as an additional qualification. We organized a three-day training program on pretesting, data collection instruments, and the technique of telephone interviews to make the data enumerators and supervisors well-skilled. For collecting data through a telephone interview, the study enrolled those patients who completed 30 days after being diagnosed with COVID-19 by RT-PCR. Each patient gave informed consent and participated voluntarily in the study. The data enumerator made an appointment with each patient on the 30th day to conduct a telephone interview on the 31st day. Before taking the consent, each patient was informed about the objectives, study procedure, risks, and benefits of participation in the study. To minimize the variability among the responses obtained by different interviewers, we trained the interviewers extensively on telephone interviews and conducted shadow interviews randomly among the responses obtained by different interviewers.

A semi-structured questionnaire was used for collecting data through telephone interviews. The questionnaire comprised variables on sociodemographic characteristics (age, sex, marital status, education, occupation, place of residence, type of family, number of family members, monthly family income), health-related quality of life, symptoms of COVID-19, and comorbidities. A checklist was used to collect laboratory data of the COVID-19 patients by reviewing their medical records obtained from the central laboratory of NIPSOM. Both data collection instruments were pretested on the COVID-19 patients diagnosed in June 2020, and accordingly, necessary corrections and modifications were performed for its finalization.

To ensure the validity of the data, each telephone interview was recorded using a digital recorder along with filling up the questionnaire on paper. To ensure the quality, consistency, and relevancy of the data, we explained data collection instruments and procedures in detail to the data enumerators and supervisors. Detailed discussion was also done on each question in the questionnaire to enable them to collect data consistently following the same approach and sequence. The supervisors supervised data enumerators during data collection, randomly selected some participants, and interviewed them to verify their responses. Moreover, the investigators examined the variability among the responses obtained by different interviewers. To ensure the quality of data and precision of data analysis, we performed double entry of data. To minimize the recall bias, data on symptoms and other relevant attributes were retrieved by reviewing the medical records of the patients, which were filled up by the trained physician during the diagnosis process. We also posed probing questions during data collection through telephone interviews to verify the response to minimize recall bias.

## Psychometric properties of the questionnaire

HRQOL was assessed via CDC's Core and Optional HRQOL Modules. We used thirteen out of fourteen items from CDC HRQOL-14 for measuring the quality of life of COVID-19 patients. We excluded question 2 of the activity limitation module from our study because it is applicable for the estimation of the overall HRQOL of an individual considering all the health problems through future follow-up. But the present study conducted the assessment of the quality-of-life changes caused only by COVID-19at a specific time (one month after diagnosis). Among 13 items, four items are part of the Core HRQOL module, and the remaining nine questions are part of a 10-item Optional Module which measures more detailed aspects of HRQOL. Question 1 assesses self-rated general health with five responses ranging from excellent to poor. For statistical analysis, responses were grouped into two categories considering the grouping done in another study conducted in Pennsylvania [14]. These categories were named as: i) 'good health', which included 'excellent, very good, or good' responses and ii) 'not so good health', which included 'fair or poor' responses. In the remaining three items of the Healthy Days Core Module (duration of physical illness, duration of mental illness, and duration of usual activity limitation) scores ranged from 0 to 30 days, with higher scores indicating poor perceived health. Unhealthy days were an estimate of the overall number of days during the previous 30 days when the respondents felt that either their physical or mental health was not good. To obtain an estimate of the overall unhealthy days of a patient, responses to questions 2 (duration of physically unhealthy days) and question 3 (duration of mentally unhealthy days) were added together, with a logical maximum of 30 unhealthy days. Eight additional items (presence of a health-related activity limitation, needing help with personal care, needing help with routine care, and five questions of healthy days' symptoms module) were treated as binary (yes/no) variables. The remainder of the HRQOL items were treated as continuous variables, with a minimum score of 0 to a maximum score of 30. The summary index of unhealthy days and healthy days was measured by the standard procedure described along with the questionnaire [15].

Though the CDC HRQOL-14 measures were initially developed for use in the U.S. English-speaking population, the concepts assessed by the measures are believed to be universal and are therefore capable of being adapted for use in other cultures and languages [11]. As the tool was not used previously in the Bangladeshi population, it was first translated into Bangla and then back translated into English. Followed by pre-testing was done to ensure the validity and reliability of the data collection instrument. Analysis revealed Cronbach's alpha reliabilities for the three modules of the CDC HRQOL-14; healthy days' core module, activity limitations module, and healthy days' symptoms module as 0.86, 0.81, and 0.76, respectively.

## Statistical analysis

Data were analyzed using SPSS STATISTICS (Version 25.0, IBM Statistical Product and Service Solutions, Armonk, NY, USA). A Chi-square Goodness-of-Fit test was used to assess the representativeness of the sample. The normality of the variables was also tested with the Shapiro-Wilk test/Kolmogorov-Smirnov tests of Normality. Continuous data were portrayed in the form of mean and standard deviation. Categorical data were depicted as frequency and percentages. Descriptive statistics estimated mean, standard deviation, and frequency while inferential statistics included chi-square test, independent sample t-test, and F-test. We used the chi-square test to compare the significant difference between two categorical variables. The t-test was used to find a significant difference in the means of two quantitative variables, and the F-test was used to find a significant difference in the means of three or more quantitative variables. Significant F-test results were further analyzed by Post-Hoc test. To identify the

confounders, we developed an adjusted logistic model using all the significant categorical variables and calculated the adjusted odds ratio (AOR) by logistic regression analysis. A p-value <0.05 was considered significant with 95% confidence interval (CI). All the statistical tests were two-sided and performed at a significance level of $\alpha$ = 0.05.

### Ethical considerations

We obtained ethical approval from the Institutional Review Board (IRB) of the National Institute of Preventive and Social Medicine (NIPSOM), Mohakhali, Dhaka-1212, Bangladesh (Ref. No. NIPSOM/IRB/2020/5, Date: 17/05/2020). Participation of the COVID-19 patients was voluntary, and they had the freedom to withdraw their consent at any stage of the study. Before data collection, informed verbal consent was obtained from each participant over the telephone after describing the study procedure and ethical issues. With the permission of each participant, the informed consent was also recorded using an audio recorder. During data collection, the confidentiality of data and privacy of the patients were maintained strictly. We used the collected data anonymously only for the current study.

### Results

Out of 1342 COVID-19 patients, 1204 (89.7%) patients finally participated in the study. Among 1342 patients, 5.1% had a wrong contact number documented, 2.1% didn't attend the phone calls, 1.9% were unwilling to participate, and 1.2% had an incomplete interview (Fig 1).

We used a Chi-square Goodness-of-Fit test to assess whether the proportion of males and females reported in the sample of the present study (Males = 72.3% and Females = 27.3%) was different than the reference population (Males = 69.86% and Females = 30.14%) [13]. The test revealed not statistically significant ($\chi2$ = 3.523, $\rho$>0.05) difference, which indicated that the study population was representative of the general population in Bangladesh.

In the present study, the majority (49.3%) of the patients were in the age group 30–49 years followed by 20.8% belonged to the age group 20–29 years, and the mean (SD) age of the patients was 41.63 (±13.53) years. Around three-fourths (72.3%) of patients were males, and the rest 27.7% were females. Of all, 82.8% were currently married, 25.1% completed their graduate-level education, and 49.6% were service holders. The urban and rural distribution of participants was almost equal (49.8% vs. 50.2%). More than two-thirds (67.5%) were from a nuclear family. The majority (48.3%) of the patients had a monthly family income between US $ 250 and US$ 595.2, and the median (IQR) of the monthly family income was US$ 297.6 (214.3–476.2) (Table 1).

Concerning symptoms of COVID-19, the majority (86.0%) of the patients had a fever. Other symptoms included cough (59.0%), sore throat (41.0%), anosmia (38.2%), shortness of breath (26.0%), and diarrhea (16.1%) (Fig 2).

Out of all 1204 patients, 35.5% of COVID-19 patients had comorbidity. Regarding the type of comorbidity, 55.6% of the patients had hypertension, and another 55.6% had diabetes mellitus. Another 16.4% of the patients had ischemic heart disease, and 12.4% had lung disease. Only 4.2% of the patients had chronic liver disease, and 2.8% had chronic kidney disease (Fig 3).

The general health condition was good in 70.1% of the patients, and it was 'not so good' in 29.8% of the patients. The majority (87.0%) of the COVID-19 patients needed help with personal care, and 47.8% required assistance with routine needs. The mean (±SD) duration of physical illness of the patients was 9.83(±7.09) days, mental illness was 7.97(±8.12) days, and the overall duration of unhealthy days was 16.27 (±9.83) days. The mean (±SD) duration of health-related limited activity was 11.39(±9.29) days. The majority (68.6%) of the patients

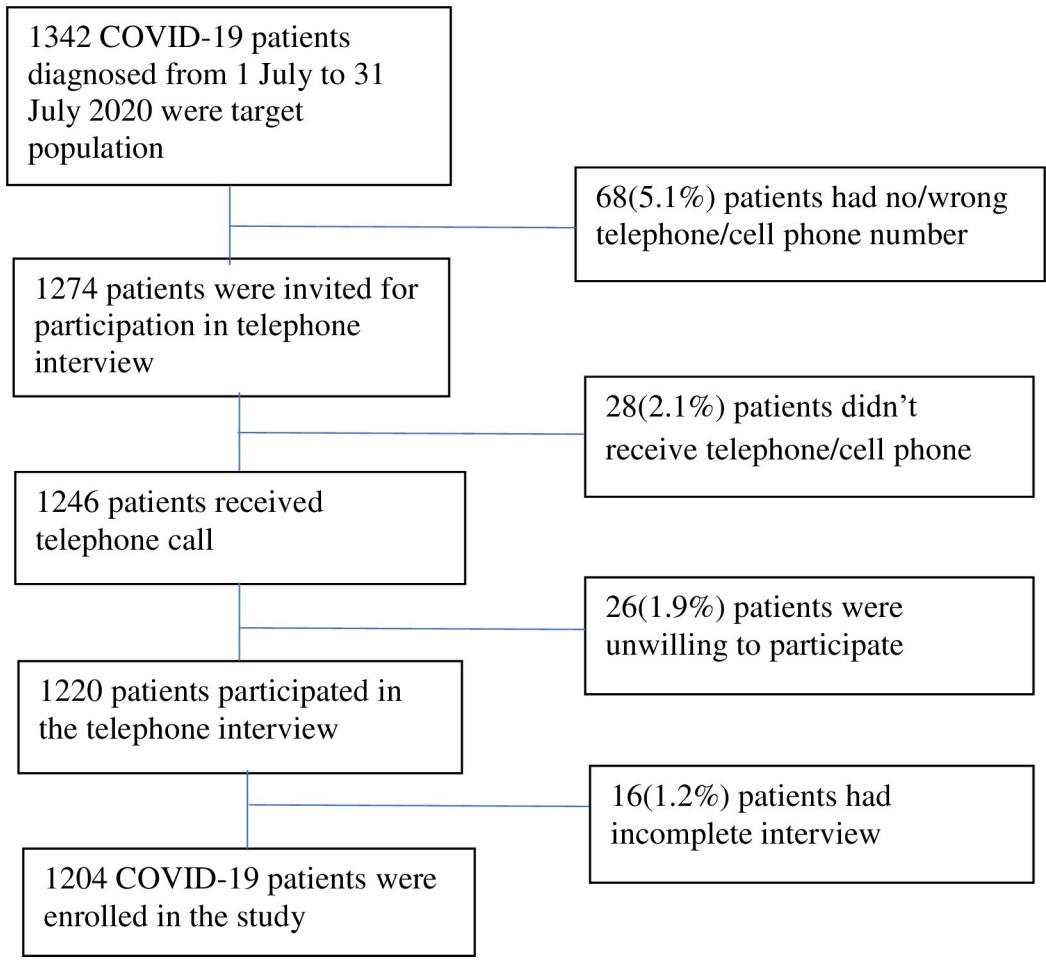

**Fig 1. Flowchart of the study participants (COVID-19 patients).**

reported both physically and mentally unhealthy days, 20.0% reported only physically unhealthy days, and 3.9% patients reported only mentally unhealthy days (Table 2).

The risk of 'not so good' general health condition was 1.57 times higher among female patients than male patients (AOR = 1.565, CI = 1.01–2.42), 1.78 times higher among service holders than homemakers (AOR = 1.784, CI = 1.04–3.06), 1.32 times higher among patients who lived in a joint family than nuclear family (AOR = 1.324, CI = 1.00–1.75), 32.87 times higher among patients having a symptom of COVID-19 than patients without symptom (AOR = 32.871, CI = 8.06–134.0), and 1.7 times higher among patients with comorbidity than patients without comorbidity (AOR = 1.700, CI = 1.26–2.29). The risk of mental distress was 1.59 times higher among female patients than males (AOR = 1.593, CI = 1.03–2.46) and 4.89 times higher among patients having a symptom of COVID-19 than patients without symptoms (AOR = 4.887, CI = 2.58–9.24). The risk of help needed with personal care was 3.59 times higher among patients in the age group of 30–49 years (AOR = 3.586, CI = 1.11–11.60) and 7.09 times higher among patients in the age group of 50–59 years (AOR = 7.088, CI = 1.13–44.56) than the selected reference group of age (20–29 years). The risk of help needed with routine needs of the patients was 2.04 times higher among urban residents than rural (AOR = 2.038, CI = 1.58–2.64), 2.80 times higher among agricultural workers and day laborers than homemakers (AOR = 2.802, CI = 1.05–7.45), 2.54 times higher among patients having a

**Table 1. Distribution of the COVID-19 patients by background characteristics (n = 1204).**

| Background characteristic | | Frequency (%) |
|---|---|---|
| **Age (Years)** | 20–29 | 251(20.8) |
| | 30–49 | 593(49.3) |
| | 50–59 | 227(18.9) |
| | 60–90 | 133(11.0) |
| | Mean(±SD) | 41.63 (±13.53) |
| **Sex** | Male | 871(72.3) |
| | Female | 333(27.7) |
| **Marital status** | Unmarried | 173(14.4) |
| | Currently married | 997(82.8) |
| | Ever married[a] | 34(2.8) |
| **Education** | Illiterate and primary | 310(25.7) |
| | Secondary | 377(31.3) |
| | Higher secondary | 215(17.9) |
| | Graduation and above | 302(25.1) |
| **Occupation** | Homemaker | 237(19.7) |
| | Business | 224(18.6) |
| | Service | 597(49.6) |
| | Retired & unemployed | 122(10.1) |
| | Agriculture worker & day labor | 24(2.0) |
| **Place of residence** | Rural | 599(49.8) |
| | Urban | 605(50.2) |
| **Type of family** | Nuclear | 813(67.5) |
| | Joint | 391(32.5) |
| **Number of family member** | 1–4 | 526(43.7) |
| | 5–9 | 598(49.7) |
| | 10–15 | 80(6.6) |
| | Mean(±SD) | 5.27 (±2.43) |
| **Monthly family income (US$[b])** | 59.5–238.1 | 502(41.7) |
| | 238.2–595.2 | 581(48.3) |
| | 595.3–2381.0 | 121(10.0) |
| | Mean (±SD) | 383.7(±307.1) |
| | Median (IQR) | 297.6(214.3–476.2) |
| **Presence of symptom** | Yes | 1067(88.6) |
| | No | 137(11.4) |
| **Presence of comorbidity** | Yes | 428(35.5) |
| | No | 776(64.5) |

SD = Standard deviation; IQR = Inter quartile range

[a]Ever married = Widowed / Widower / Separated / Divorced

[b]US$ = United States Dollar (Currency of United States of America)

symptom of COVID-19 than patients without symptom (AOR = 2.537, CI = 1.68–3.84), and 1.50 times higher among patients with comorbidity than patients without comorbidity (AOR = 1.501, CI = 1.14–1.98) (Table 3).

The mean duration of healthy days was significantly (p<0.05) lower in the ever-married group (11.24 days) as well as among the patients having a symptom (11.96 days) and comorbidity (12.23 days). The mean duration of usual activity limitation was significantly (p<0.05)

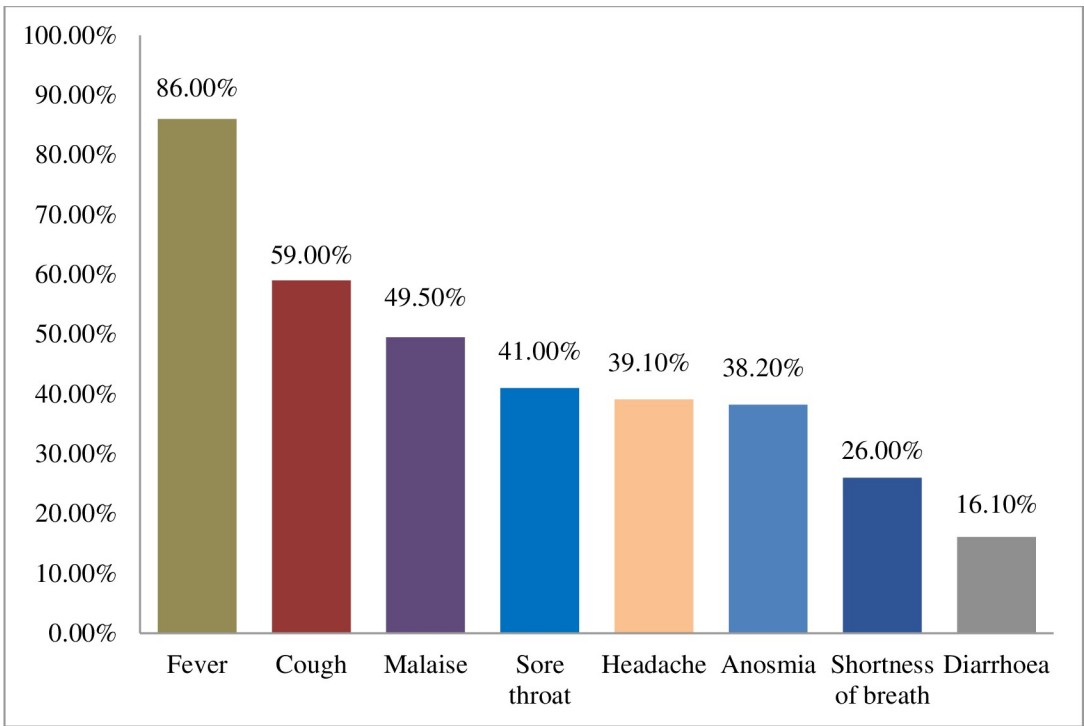

**Fig 2. Distribution of the symptoms of the COVID-19 patients.**

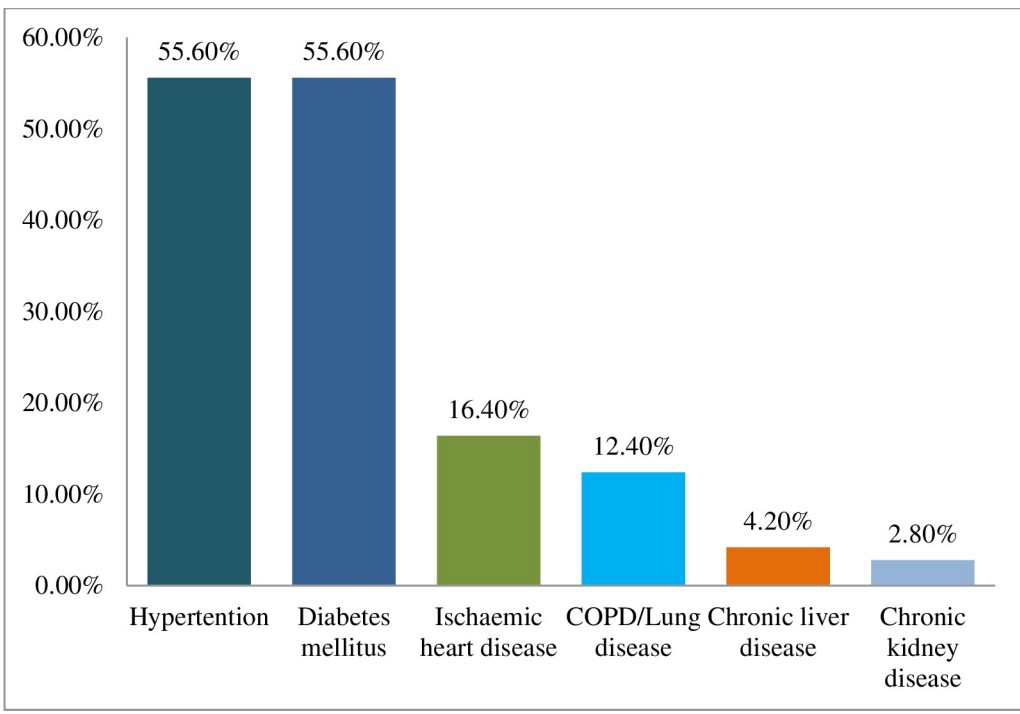

**Fig 3. Distribution of the COVID-19 patients by comorbidities (n = 427).**

**Table 2. Distribution of different items of health-related quality of life of the COVID-19 patients (based on CDC HRQOL-14).**

| Items of healthy days core module | | |
|---|---|---|
| Patients' general health condition, n (%) | Good | 845(70.1) |
| | Not so good | 359(29.8) |
| Duration of physical illness (Mean±SD days) | | 9.83(±7.09) |
| Duration of mental illness (Mean±SD days) | | 7.97(±8.12) |
| Duration of usual activity limitation (Mean±SD days) | | 7.04(±5.53) |
| **Items of activity limitations module** | | |
| Duration of health-related limited activity (Mean±SD days) | | 11.39(±9.29) |
| Need help for personal care, n (%) | Yes | 1047(87.0) |
| | No | 157(13.0) |
| Need help for routine needs, n (%) | Yes | 576(47.8) |
| | No | 628(52.2) |
| **Items of healthy days symptoms module** | | |
| Duration of hard to do usual activities due to pain (Mean±SD days) | | 3.88(±7.01) |
| Duration of feeling sad or depressed (Mean±SD days) | | 7.04(±8.07) |
| Duration of feeling worried (Mean±SD days) | | 6.77(±8.03) |
| Duration of not getting enough rest (Mean±SD days) | | 4.38(±6.79) |
| Duration of feeling healthy (Mean±SD days) | | 14.83(±9.49) |
| **Overall duration of unhealthy days (Mean±SD days)[a]** | | **16.27(±9.83)** |
| **Number of patients with physically and mentally unhealthy days:** | | |
| Only physically unhealthy days (%) | | 241(20.0) |
| Only mentally unhealthy days (%) | | 47(3.9) |
| Both physically and mentally unhealthy days (%) | | 826(68.6) |
| None (%) | | 90(7.5) |

n = Number; % = Percentage; SD = Standard deviation

[a]Unhealthy days were calculated by using the standard assessment procedure of HRQOL designed by CDC [11].

higher in the age group 50–59 years (7.88 days), patients having symptoms (7.94 days), and comorbidity (7.88 days). The mean duration of health-related limited activity was significantly (p<0.05) higher in the males (11.75 days) and patients having symptoms (12.23 days) (Table 4).

The mean duration of 'feeling pain' was significantly (p<0.05) associated with the presence of symptoms (4.33 days) and comorbidity (5.28 days). The mean duration of 'feeling sad, blue, or depressed' was significantly (p<0.05) higher in ever-married patients (10.47 days), patients having symptoms (7.58 days), and comorbidity (8.04 days). The mean duration of 'feeling worried' was significantly (p<0.05) associated with gender (male = 6.48 days and female = 7.53 days) and having a symptom (7.26 days). The mean duration of 'feeling worried' was significantly (p<0.05) higher among ever-married patients (10.12 days) (Table 5).

The mean duration of 'not getting enough rest' was significantly (p<0.05) higher in females (5.26 days), having a symptom (4.69 days) and comorbidity (5.02 days). However, the mean duration of 'feeling very healthy' was significantly (p<0.05) lower in the age group 50–59 years (13.24 days). The mean duration of 'feeling very healthy' was also significantly (p<0.05) lower in the patients having symptoms (13.51 days) and comorbidity (13.0 days) (Table 5).

**Table 3. Logistic regression analysis of items of HRQOL by background and clinical characteristics of the COVID-19 patients (n = 1204).**

| Background and clinical characteristic | | Not so good general health condition AOR (95%CI) | Mental distress AOR (95%CI) | Need help with personal care AOR (95%CI) | Need help with routine needs AOR (95%CI) |
|---|---|---|---|---|---|
| **Age (Years)** | 20–29 | Reference | Reference | Reference | Reference |
| | 30–49 | 0.923(0.59–1.43) | 0.775(0.50–1.19) | 3.586(1.11–11.60)* | 1.272(0.85–1.91) |
| | 50–59 | 1.050(0.61–1.80) | 0.747(0.44–1.28) | 7.088(1.13–44.56)* | 1.386(0.84–2.28) |
| | 60–90 | 1.103(0.60–2.02) | 0.877(0.48–1.60) | 3.323(0.05–24.05) | 1.020(0.58–1.80) |
| **Sex** | Male | Reference | Reference | Reference | Reference |
| | Female | 1.565(1.01–2.42)* | 1.593(1.03–2.46)* | 0.314(0.07–1.52) | 0.982(0.65–1.49) |
| **Marital status** | Unmarried | Reference | Reference | Reference | Reference |
| | Currently married | 1.026(0.62–1.69) | 1.413(0.85–2.34) | 0.646(0.12–3.62) | 0.925(0.59–1.46) |
| | Ever married[a] | 1.001(0.39–2.60) | 1.547(0.60–3.98) | 0.355(0.02–6.29) | 0.826(0.34–2.04) |
| **Education** | Up to primary | Reference | Reference | Reference | Reference |
| | Secondary | 1.032(0.71–1.50) | 0.847(0.58–1.23) | 1.517(0.41–5.59) | 0.671(0.48–0.95)* |
| | Higher secondary | 0.942(0.60–1.49) | 0.940(0.60–1.62) | 0.984(0.20–4.87) | 0.952(0.63–1.45) |
| | Graduation and above | 1.013(0.64–1.60) | 1.033(0.66–1.62) | 0.521(1.12–2.30) | 1.106(0.73–1.67) |
| **Occupation** | Homemaker | Reference | Reference | Reference | Reference |
| | Business | 1.194(0.66–2.17) | 1.093(0.61–1.97) | 0.244(0.04–1.60) | 1.262(0.73–2.19) |
| | Service | 1.784(1.04–3.06)* | 1.301(0.77–2.20) | 1.79(0.34–9.42) | 1.258(0.76–2.08) |
| | Retired / unemployed | 1.778(0.95–3.33) | 1.342(0.72–2.50) | - | 1.506(0.84–2.71) |
| | Agriculture & day labor | 2.082(0.74–5.88) | 1.039(0.35–3.10) | - | 2.802(1.05–7.45)* |
| **Place of residence** | Rural | Reference | Reference | Reference | Reference |
| | Urban | 1.069(0.81–1.42) | 1.049(0.79–1.39) | 1.200(0.45–3.23) | 2.038(1.58–2.64)* |
| **Type of family** | Nuclear | Reference | Reference | Reference | Reference |
| | Joint | 1.324(1.00–1.75)* | 1.221(0.92–1.61) | 1.294(0.48–3.49) | 1.043(0.80–1.35) |
| **Monthly family income (US$[b])** | 59.5–238.1 | Reference | Reference | Reference | Reference |
| | 238.2–595.2 | 0.735(0.55–0.99)* | 0.814(0.61–1.09) | 0.840(0.30–2.35) | 1.153(0.88–1.51) |
| | 595.3–2381.0 | 0.718(0.45–1.16) | 0.708(0.43–1.15) | 1.914(0.22–17.05) | 1.686(1.08–2.64)* |
| **Having symptom** | Yes | 32.871(8.06–134.0)* | 4.887(2.58–9.24)* | - | 2.537(1.68–3.84)* |
| | No | Reference | Reference | Reference | Reference |
| **Having comorbidity** | Yes | 1.700(1.26–2.29)* | 1.260(0.93–1.70) | 1.627(0.53–5.02) | 1.501(1.14–1.98)* |
| | No | Reference | Reference | Reference | Reference |

AOR = Adjusted Odds Ratio; CI = Confidence interval

* Significant at p = 0.05 level with 95% CI

[a]Ever married = Widowed / Widower / Separated / Divorced

[b]US$ = United States Dollar

## Discussion

This cross-sectional study appraised the HRQOL of adult COVID-19 patients in the context of Bangladesh. The study enrolled COVID-19 patients who had their diagnosis confirmed by the RT-PCR test following the case definition for COVID-19 specified in the national guideline of Bangladesh [16]. The current study was an impressive initiative to assess the health-related quality of COVID-19 patients using CDC-HRQOL on national and international platforms because it used the scale to measure HRQOL of COVID-19 patients considering both physical and mental health along with limitations of daily health activities. The study findings preserve potential policy implications to contrive realistic guidelines for improving the quality of life

**Table 4. Mean±SD duration of selected items of HRQOL by background and clinical attributes of the COVID-9 patients (n = 1204).**

| Attributes | | Duration of healthy days (Mean±SD) | p-value | Duration of usual activity limitation (Mean±SD) | p-value | Duration of health–related limited activity (Mean±SD) | p-value |
|---|---|---|---|---|---|---|---|
| **Age (Years)** | 20–29 | 14.88(10.13)[a] | | 6.12(5.00)[a] | | 11.09(9.56)[a] | |
| | 30–49 | 13.88(9.69) | 0.524 | 7.06(5.64) | 0.105 | 11.80(9.33) | 0.741 |
| | 50–59 | 12.96(9.62) | 0.143 | 7.88(5.85) | 0.003 | 11.34(8.98) | 0.920 |
| | 60–90 | 12.19(10.01) | 0.052 | 7.26(5.26) | 0.211 | 10.17(9.06) | 0.261 |
| **Sex** | Male | 13.88(9.71) | 0.391 | 7.14(5.35) | 0.323 | 11.75(9.31) | 0.028 |
| | Female | 13.33(10.13) | | 6.78(5.99) | | 10.44(9.18) | |
| **Marital status** | Unmarried | 15.61(10.16)[a] | | 6.34(5.41)[a] | | 11.86(9.75)[a] | |
| | Currently married | 13.48(9.76) | 0.023 | 7.19(5.59) | 0.148 | 11.38(9.26) | 0.810 |
| | Ever married[b] | 11.24(9.03) | 0.046 | 6.09(4.25) | 0.968 | 9.12(7.31) | 0.258 |
| **Having symptom** | Yes | 11.96(8.84) | 0.000 | 7.94(5.23) | 0.000 | 12.23(9.08) | 0.000 |
| | No | 27.45(5.25) | | 0.0(0) | | 4.82(8.20) | |
| **Having comorbidity** | Yes | 12.23(9.78) | 0.000 | 7.88(5.97) | 0.000 | 10.88(9.30) | 0.161 |
| | No | 14.55(9.76) | | 6.57(5.23) | | 11.66(9.28) | |

SD = Standard deviation

* t-test/Post-Hoc test, Significant at $\rho<0.05$ level with 95% CI

[a] Reference category

[b]Ever married = Widowed / Widower / Separated / Divorced

and health condition of COVID-19 patients with effective clinical management. The study findings based on the experiences of COVID-19 patients and considering their health conditions, illnesses (both physical and mental), and usual activity limitations might help to update the national guideline on clinical management for providing need-based healthcare to COVID-19 patients. The study also conserves enormous academic and research implications to track the clinical course of COVID-19 and its impacts on the physical and mental health of the patients. As the Chi-square Goodness-of-Fit test indicated that the sample was representative of the national population, the findings could contribute to reforming and redesigning the healthcare delivery system for COVID-19 patients in local and relevant contexts.

## Background characteristics

In the present study, around three-fourths of the COVID-19 patients were males, and the rest were females. This finding indicates that males were at a higher risk of being infected by COVID-19 than females. In this respect, other studies conducted in China [17] and Vietnam [8] revealed reverse findings, where females were affected more than males. It may be due to the divergence of the socio-cultural context of Bangladesh, which is different from those countries. Based on the local social and cultural frame, males are more involved in outdoor activities and more exposed to social gatherings than females. So, the chance of exposure and being affected is higher in males than females. Around half of the COVID-19 patients were young adults (age group 30–49 years), and only 11.0% of the patients were elderly (aged 60–90 years). The demographic profile of Bangladesh depicts that 40.07% of the population belongs to the age group of 25–54 years, and only 6.42% belongs to the age group >60 years [1]. In this regard, we can claim that the age distribution of the patients is in alignment with the national age distribution.

An almost equal number of the COVID-19 patients reported from the urban (50.2%) and rural (49.8%) communities. The study of Islam MZ et al., [1] identified a different finding in

**Table 5. Mean(±SD) duration of selected items of HRQOL by background and clinical attributes of the COVID-19 patients (n = 1204).**

| Attributes | | Feeling pain (Mean ±SD) | p-value | Feeling sad/ blue/depressed (Mean ±SD) | p-value | Feeling worried (Mean ±SD) | p-value |
|---|---|---|---|---|---|---|---|
| **Age (Years)** | 20–29 | 2.94(6.25)[a] | Reference | 6.39(7.59)[a] | Reference | 6.20(7.41)[a] | Reference |
| | 30–49 | 3.92(6.97) | 0.246 | 6.98(7.90) | 0.763 | 6.77(7.89) | 0.778 |
| | 50–59 | 4.59(7.27) | 0.051 | 7.20(8.30) | 0.690 | 6.90(8.36) | 0.774 |
| | 60–90 | 4.28(7.89) | 0.284 | 8.28(9.18) | 0.129 | 7.59(9.14) | 0.373 |
| **Sex** | Male | 3.69(6.67)[b] | 0.114 | 6.86(7.95)[b] | 0.212 | 6.48(7.90)[b] | 0.041 |
| | Female | 4.40(7.80) | | 7.51(8.37) | | 7.53(8.33) | |
| **Marital status** | Unmarried | 2.83(6.38)[a] | Reference | 5.92(7.04)[a] | Reference | 5.57(6.80)[a] | Reference |
| | Currently married* | 4.08(7.13) | 0.077 | 7.12(8.18) | 0.165 | 6.86(8.16) | 0.124 |
| | Ever married | 3.38(5.83) | 0.908 | 10.47(8.85) | 0.007 | 10.12(9.06) | 0.007 |
| **Having symptom** | Yes | 4.33(7.28)[b] | 0.000 | 7.58(8.19)[b] | 0.000 | 7.26(8.20)[b] | 0.000 |
| | No | 0.39(2.17) | | 2.86(5.56) | | 2.91(5.17) | |
| **Having comorbidity** | Yes | 5.28(8.29)[b] | 0.000 | 8.04(8.51)[b] | 0.001 | 7.36(8.34)[b] | 0.059 |
| | No | 3.12(6.06) | | 6.49(7.77) | | 6.44(7.84) | |
| **Attributes** | | Not getting enough rest (Mean ±SD days) | | p-value | | Feeling very healthy (Mean±SD days) | p-value |
| **Age (Years)** | 20–29 | 4.16(6.61)[a] | | Reference | | 16.07(9.63)[a] | Reference |
| | 30–49 | 4.66(7.13) | | 0.770 | | 15.14(9.25) | 0.560 |
| | 50–59 | 3.83(5.86) | | 0.951 | | 13.24(9.37)* | 0.006 |
| | 60–90 | 4.53(7.05) | | 0.959 | | 13.80(10.08) | 0.115 |
| **Sex** | Male | 4.05(6.35)[b] | | 0.006 | | 15.03(9.43)[b] | 0.236 |
| | Female | 5.26(7.76)[b] | | | | 14.03(9.62)[b] | |
| **Marital status** | Unmarried | 3.64(6.22)[a] | | Reference | | 16.37(9.60)[a] | Reference |
| | Currently married | 4.46(6.83) | | 0.299 | | 14.64(9.48) | 0.068 |
| | Ever married[c] | 5.82(8.12) | | 0.198 | | 12.53(8.25) | 0.078 |
| **Having symptom** | Yes | 4.69(6.89)[b] | | 0.000 | | 13.51(8.74)[b] | 0.000 |
| | No | 2.01(5.43) | | | | 25.05(8.84) | |
| **Having comorbidity** | Yes | 5.02(7.32)[b] | | 0.015 | | 13.0(9.86)[b] | 0.000 |
| | No | 4.03(6.45) | | | | 15.84(9.13) | |

SD = Standard deviation; CI = Confidence interval; Significant at p = 0.05 level with 95% CI

[a]Post-Hoc test

[b]t-test [c]Ever married = Widowed / Widower / Separated / Divorced

Bangladesh where reported patients were more in number from the urban communities. Though both the studies were carried out in the same country, a different result may be due to differences in the study periods. The former study was conducted earlier compared to the current one. At the beginning of the COVID-19 pandemic, limited diagnostic facilities were available in the urban setting of Bangladesh. Moreover, rural people of the country were less aware of the COVID-19 diagnosis. As a result, the former study found more patients reported from the urban setting than in the current study.

In respect of occupation, nearly half of the patients were service holders. In this regard, the study conducted by Islam MZ et al., [1] also identified most of the patients as service holders, and it was relatively lower (32.5%) than in the current study. It is evident that a significant portion of the population of Bangladesh is engaged in various occupational activities, and they must move frequently within the community, which exposes them to disease infection. As a result, the service holders are more vulnerable to getting an infection of COVID-19 than other

occupational groups. We can mention that during the early stage of the pandemic, there was a restriction on population movement due to lockdown, shutdown, quarantine, and panic about the disease. It could be a valid reason for comparatively fewer service holders in the former than in the current study.

### Symptoms and comorbidities of COVID-19 patients

Due to poor diagnostic facilities and limited RT-PCR testing capacity, asymptomatic individuals in a developing nation like Bangladesh were less likely to be investigated for disease screening. Moreover, due to ignorance and superstition, a significant proportion of asymptomatic patients didn't participate in the investigation procedure of the disease. Lack of contact tracing and community-based screening initiatives in a setting with limited resources may also be responsible for the detection of fewer asymptomatic patients. The study found that 11.4% of asymptomatic patients attended the health facilities to exclude COVID-19 followed by exposure history, travel history, and treatment of diverse ailments. The present study portrayed that more than one-third (35.5%) of the COVID-19 patients had different comorbidities like hypertension, diabetes mellitus, ischemic heart disease, chronic lung, kidney, and liver diseases. The former study by Islam MZ et al., [1] detected nearly the same proportion (33.9%) of COVID-19 patients with similar comorbidities. Due to the higher risk of worse morbidity and mortality consequences of COVID-19 among the comorbid patients, they were referred to different health facilities to exclude COVID-19.

### Different items of HRQOL of the COVID-19 patients and associated factors

We used laboratory records to retrieve the data from those COVID-19 patients who completed their 30 days' duration of illness after being RT-PCR positive. We obtained information on patients' general health condition, mental and physical illness, health-related activity limitations, needing help with personal and routine care, and healthy days' symptoms module. We evaluated and analyzed all items separately.

At the end of 30 days after diagnosis by RT-PCR test, the general health condition was 'good' in 70.1% of patients, and it was 'not so good' in 29.8% of patients. 'Not so good' health condition was significantly associated with female gender (AOR = 1.565, CI = 1.01–2.42), joint family (AOR = 1.324, CI = 1.00–1.75), having a symptom of COVID-19 (AOR = 32.871, CI = 8.06–134.0), and comorbidity (AOR = 1.700, CI = 1.26–2.29). 'Not so good' health condition also increased with increasing age. The symptoms of COVID-19 aggravate the morbidity and illness severity of the patients, and in turn, interfere with their general health condition. It is a fact that females and elderly patients conserve relatively lower body immunity to protect against the illness progression and its clinical severity. As a result, their duration of suffering lasts longer than the males and younger adults.

The estimated mean duration of physical illness was around ten days and eight days for mental illness. The mean duration of the usual activity limitation was around seven days and it was significantly higher in the age group 50–59 years. The presence of comorbidity poses an incremental effect on the worse progression of the disease and impairs the health condition of the patients. In the present study, patients' need for help with routine needs was also significantly higher among urban patients having a symptom and comorbidity. The duration of limited health-related activity was significantly higher in males and patients having symptoms and comorbidity. The duration of healthy days was significantly lower among patients with symptoms and comorbidities. Usual activity limitation and limited health-related activity were significantly higher among patients having symptoms. Coexisting clinical symptoms and

comorbidities deteriorate the physical and mental well-being of the patients, which decreases their ability to perform daily activities.

The mean duration of feeling pain was significantly higher in patients having symptoms and comorbidity. The mean duration of feeling sad, blue, or depressed was higher in the ever-married patients and those having symptoms and comorbidity. A study conducted in China revealed a higher risk of pain/discomfort and anxiety/depression among older people having a chronic disease and lower income [17]. On the contrary, the mean duration of feeling worried was significantly higher in females, ever-married patients, and those having symptoms. The mean duration of feeling very healthy was lower in the older adults (50–59 years) and those having symptoms and comorbidity. The present study revealed a significantly higher duration of symptoms in females, ever married, and patients with clinical manifestation and comorbidity. These findings entice the attention of the health policymakers and healthcare managers to invent prioritized medical measures for those high-risk COVID-19 patients to improve their HRQOL and general health condition.

The current study concerns a limitation in collecting some relevant information like severity of COVID-19 infection, treatment options, presence of any other short-term illness during the one month, etc. We could not identify and manage the heterogeneity in the COVID-19 severity among the sample because the severity could not be measured through telephone interviews and reviewing medical records having scarcity of relevant information. It is a fact that any short-term disease could impair the HRQOL during the 30 days of illness period and could confound the effect of COVID-19. We asked the patients about any short-term illness, but they gave no significant response. We did not perform any clinical examinations and investigations of the patients due to their sporadic distribution throughout the country. Telephone interviews and inadequate information in the medical records were also a few reasons behind it. This cross-sectional study design only provided data on the HRQOL of COVID-19 patients after 30 days of being diagnosed, and no comparison was possible due to the lack of a control group and any previous baseline data. So, we could not determine whether the HRQOL of the patients was stabilized or returned to baseline after 30 days. We only assessed the HRQOL of the COVID-19 patients for one month since diagnosis (whether it was stabilized/returned to baseline or not) following the specific guidelines of the CDC HRQOL-14 questionnaire. Although we considered a shorter period of illness to prevent recall bias, measuring the effects of COVID-19 on the HRQOL of the patients couldn't be impartial. The study considered clinical symptoms, treatment, comorbidities, and socioeconomic features as crucial determinants of HRQOL. The sample size was adequate for exploring the patients' experiences and providing substantial evidence and directions for policy-making and future analytical studies.

### Advantages of CDC HRQOL-14 questionnaire

Firstly, there is a policy value of estimating the burden of disease or disability in days, months, or years because it provides concrete measures that can be understood and used by legislators, policymakers, and healthcare managers to devise effective interventions in the preventive and clinical management of COVID-19. Secondly, the healthy days' measures permit a wide range of options for statistical analysis because of their continuous, cardinal, and bounded (Range = 0 to 30) mathematical properties. Moreover, the quantitative approach of CDC HRQOL over other tools offers ample scope to measure the duration of physical and mental illness. Accordingly, it contributes to developing a vibrant clinical guideline for the effective management of diverse communicable and non-communicable diseases. The core CDC HRQOL-14 measures are the briefest validated set of generic HRQOL measures with a clear

and explicit definition of HRQOL, and a transparent summary measure. The CDC HRQOL-14 measure is the most cost-effective and less burdensome among all HRQOL measures [15]. The CDC HRQOL-14 questionnaire is designed in such a manner that it can estimate the overall number of days during the previous 30 days when the respondents felt that either their physical or mental health was not good.

The CDC HRQOL-14 instrument can calculate the HRQOL of the patients for 30 days, whether it has returned to baseline/stabilized or not [11]. So, the instrument was appropriate for the present study to assess HRQOL of COVID-19 patients comprising physical and mental health for 30 days after diagnosis. One of the additional advantages of the CDC HRQOL-14 questionnaire is that the whole interview could be completed over the telephone, a data collection technique that is profoundly beneficial during this COVID-19 pandemic [12].

## Conclusion

The present study measured the substantial impact of COVID-19 on HRQOL by the CDC HRQOL-14 questionnaire. The average duration of overall unhealthy days was 16.27 days. The study identified the factors that have an impact on HRQOL of COVID-19 patients, and it comprised female gender, having symptoms of COVID-19, and comorbidity. The findings on the duration of physical and mental illness and the factors associated with HRQOL could contribute to updating the national guideline for effective clinical management of COVID-19 patients. The study results could also contribute to making a follow-up schedule for the COVID-19 survivors to evaluate their physical and mental well-being. The study invites special attention to devising clinical measures and program interventions considering physical illness, mental illness, and the limitation of daily activities of the COVID-19 patients.

## Supporting information

**S1 Dataset. De-identified original dataset (SPSS).**
(SAV)

## Acknowledgments

All authors would like to acknowledge the laboratory authority, medical technologists, and supporting staff for their continuing and unconditional support in sharing laboratory records of the diagnosed patients. We are pleased to forward our sincere appreciation to the COVID-19 patients and their families for their unrestricted cooperation in data collection through a telephone interview.

## Author Contributions

**Conceptualization:** Md. Ziaul Islam, Sharmin Farjana.

**Data curation:** Md. Ziaul Islam, Baizid Khoorshid Riaz, Syeda Sumaiya Efa, Sharmin Farjana, Fahad Mahmood.

**Formal analysis:** Md. Ziaul Islam, Syeda Sumaiya Efa, Fahad Mahmood.

**Investigation:** Md. Ziaul Islam, Baizid Khoorshid Riaz, Syeda Sumaiya Efa, Sharmin Farjana, Fahad Mahmood.

**Methodology:** Md. Ziaul Islam, Baizid Khoorshid Riaz, Syeda Sumaiya Efa.

**Project administration:** Md. Ziaul Islam, Sharmin Farjana.

**Resources:** Sharmin Farjana, Fahad Mahmood.

**Software:** Md. Ziaul Islam.

**Supervision:** Md. Ziaul Islam, Baizid Khoorshid Riaz, Fahad Mahmood.

**Validation:** Md. Ziaul Islam, Fahad Mahmood.

**Visualization:** Md. Ziaul Islam, Sharmin Farjana.

**Writing – original draft:** Md. Ziaul Islam, Baizid Khoorshid Riaz, Syeda Sumaiya Efa, Sharmin Farjana, Fahad Mahmood.

**Writing – review & editing:** Md. Ziaul Islam, Baizid Khoorshid Riaz, Syeda Sumaiya Efa, Sharmin Farjana, Fahad Mahmood.

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
