## [Decision Letter · Decision Letter 0]

28 Jun 2022

PONE-D-22-12676Health-related quality of life of the adult COVID-19 patients following one-month illness experience since diagnosis: findings of a cross-sectional study in BangladeshPLOS ONE

Dear Dr. Islam,

Thank you for submitting your manuscript to PLOS ONE. After careful consideration, we feel that it has merit but does not fully meet PLOS ONE’s publication criteria as it currently stands. Therefore, we invite you to submit a revised version of the manuscript that addresses the points raised during the review process.

The two reviewers have given thorough inputs for improving the manuscript. Several points that should be highlighted are (i) the rationale of the study and measures being used (ii) the writing of Results and Discussions sections should be more clear and systmeatic, and (iii) copyediting of English is a must.

We look forward to receiving your revised manuscript.

Kind regards,

Fredrick Dermawan Purba, PhD

Academic Editor

PLOS ONE

Journal Requirements:

Reviewers' comments:

Reviewer's Responses to Questions

**Comments to the Author**

1. Is the manuscript technically sound, and do the data support the conclusions?

Reviewer #1: Partly

Reviewer #2: Partly

2. Has the statistical analysis been performed appropriately and rigorously? 

Reviewer #1: Yes

Reviewer #2: Yes

3. Have the authors made all data underlying the findings in their manuscript fully available?

Reviewer #1: Yes

Reviewer #2: No

4. Is the manuscript presented in an intelligible fashion and written in standard English?

Reviewer #1: No

Reviewer #2: No

5. Review Comments to the Author

Reviewer #1: The methodology needs to be written in a more scientific manner explaining sample size calculation and other criteria's. Results section is quite confusing and need to be presented in a more understandable manner. Discussion and conclusion need to be rewritten with respect to study objectives. Article needs massive English language editing.

Reviewer #2: This manuscript reported the HRQOL of COVID-19 patients in a cross-sectional sample in Bangladesh using CDC HRQOL-14 questionnaire. This is an important aspect of patient well-being, and the results should inform enhanced patient-specific support. Questions/ comments from my side are presented in the attached file.

6. PLOS authors have the option to publish the peer review history of their article (what does this mean?). If published, this will include your full peer review and any attached files.

Reviewer #1: **Yes: **madeeha malik

Reviewer #2: No

---

## [Author Response · Author response to Decision Letter 0]

23 Jul 2022

Responses to the comments of the academic editor 

1. Comment: The rationale of the study and measures being used should be highlighted

Response: Thank you for the very important comment. The ‘Rationale’ of the study and measures being used are highlighted.

2. Comment: The writing of Results and Discussions sections should be more clear and systematic

Response: Thank you for the pertinent comment. The ‘Results’ and ‘Discussion’ section are rewritten to make it more clear and systematic.

3. Comment: Copyediting of English is a must.

Response: Thank you for the vital comment. Copyediting of English is done.

4. Comment: A rebuttal letter that responds to each point raised by the academic editor and reviewer(s). You should upload this letter as a separate file labeled 'Response to Reviewers'.

Response: Thank you for the crucial comment. A separate file labeled 'Response to Reviewers' is submitted.

5. Comment: A marked-up copy of your manuscript that highlights changes made to the original version. You should upload this as a separate file labeled 'Revised Manuscript with Track Changes'.

Response: Thank you for the very valuable comment. A separate file labeled 'Revised Manuscript with Track Changes' is prepared and submitted.

6. Comment: An unmarked version of your revised paper without tracked changes. You should upload this as a separate file labeled 'Manuscript'.

Response: Thank you for the crucial comment. A separate file labeled 'Manuscript' is prepared and submitted.

7. Comment: Please ensure that your manuscript meets PLOS ONE's style requirements, including those for file naming.

Response: Thank you for the pertinent comment. The manuscript is formatted as per the requirement of PLOS ONE's style.

8. Comment: Upon re-submitting your revised manuscript, please upload your study’s minimal underlying data set as either Supporting Information files or to a stable, public repository and include the relevant URLs, DOIs, or accession numbers within your revised cover letter.

Response: Thank you for the very valuable comment. The minimal underlying dataset of the study is shared with the submission under the heading of ‘Supporting Information file’.

9. Comment: Your ethics statement should only appear in the Methods section of your manuscript. If your ethics statement is written in any section besides the Methods, please delete it from any other section.

Response: Thank you for the very important comment. Ethics statement is deleted from other sections besides the ‘Methods’ section.

10. Comment: Please include captions for your Supporting Information files at the end of your manuscript, and update any in-text citations to match accordingly.

Response: Thank you for the major comment. The captions for the supporting information file S1 Data set (De-identified Original Data set in SPSS) is included at the end of the manuscript.

Responses to comments of the reviewer 1 

1. Comment: Line 78: Please mention year along with the name of the month.

Response: Thank you for the crucial comment. Year is mentioned in the line 82.

2. Comment: The data presented in lines 79-80 may be updated.

Response: Thank you for the vital comment. Data is updated in the lines 83-84 and accordingly, the reference no. 4 is updated.

3. Comment: What data did authors use to inform that in COVID-19 patients, the duration of 30 days after the RT-PCR is appropriate to assess the effect of disease on their HRQOL (as their HRQOL is stabilised / returned to baseline)? Is there a reference or any data to back up this determination?

Response: Thank you for the vibrant comment. Health related quality of life is a multidimensional concept for examining the impact of health status on quality of life. CDC HRQOL-14 questionnaire is designed in a manner so that it can estimate the overall number of days during the previous 30 days when the respondent felt that either his or her physical or mental health was not good. The "number of days in the past 30 days" response format of the Healthy Days Measures makes them particularly well suited to respond to short-term changes in HRQOL. This instrument can calculate the HRQOL of 30 days, despite it has returned to baseline or stabilized. For this reason, it is appropriate to assess the condition of 30 days after diagnosis of COVID-19 as the disease has immediate impact on physical and mental condition of the patients.

Also the CDC HRQOL-14 is used for chronic health conditions such as arthritis, asthma, disability, depression, any emerging disease, communicable disease, emergency health event like disaster etc. and also it can be used in healthy population. The aggregated core data on ‘healthy days’ are responsive to the effect of a major disease or disaster. Details can be found in the article of Moriarty DG et al (Reference12 in the article). It is also mentioned in lines 535-536 under discussion section of the article.

4. Comment: Have all the patients been interviewed on the 30th day of obtaining positive RT-PCR? If the duration between the test and interview was not uniform, please report the median duration.

Response: Thank you for the very important comment. All the patients were interviewed on the 31st day after RT-PCR positive. On the 30th day, informed consent was taken from each patient and an appointment was taken for the next-day interview on the 31st day. We have described details in the lines 181 – 185 in the ‘Materials and methods’ section.

5. Comment: Whereas some people with COVID-19 suffer severe, even life-threatening complications, others suffer no symptoms or just mild ones. How did the authors identify and manage this heterogeneity in the sample?

Response: Thank you for the pertinent comment. Data were collected by telephone interview and it was not possible to differentiate the severity of the COVID-19 patients with verbal information as it will harvest recall bias. It could be considered as a limitation of this study which is mentioned in the lines 206-209 under the heading of ‘Data collection’ of the ‘Methods’ section and in the lines 514-517 of the ‘Discussion’ section.

6. Comment: Which all parameters / measures were used to ensure the quality, consistency, and relevancy of the data?

Response: Thank you for the vital comment. Details of all measures are included in the lines 196-208 under the heading of ‘Data collection’ in the ‘Methods’ section.

7. Comment: Please provide the reference for grouping the response of question 1 of healthy days’ core module into ‘good health’ and ‘not so good health’. 

Response: Thank you for the crucial comment. Reference is added in the lines 218-220 under the heading of ‘Psychometric properties of the questionnaire’ in the ‘Material and methods; section.

8. Comment: Line 175-177: Authors have reported that in addition to the response of question on general health (good health/ not so good health), the responses of nine additional items were treated as binary (yes/no). Shouldn’t it be eight additional items?

Response: Thank you for the vital comment. It is corrected in the line 228 under the heading of ‘Psychometric properties of the questionnaire’ in ‘the Material and methods; section.

9. Comment: Please provide details of the number of interviewers, their qualification, and their training process. Did authors investigate the variability among the responses obtained by different interviewers?

Response: Thank you for the important comment. Details are given in the lines 178-181 and 202-208 under the heading of ‘Data collection’ in the ‘Materials and methods’ section.

10. Comment: Lines 189-201 may be shifted to discussion section.

Response: Thank you for the crucial comment. The lines 189-201 describing the ‘advantages of CDC HRQOL-14 questionnaire’ are shifted to the discussion section.

11. Comment: In the method section, please write data on which sociodemographic characteristics were collected and being used in data analysis.

Response: Thank you for the vital comment. Data, on which sociodemographic characteristics were collected and being used in data analysis are listed in the lines 189-191 under the heading of data collection in the ‘Methods’ section.

12. Comment: Is there any way to assess if the characteristics of the individuals excluded from the study were similar / different from those of included (lines 228-230)?

Response: Thank you for the valuable comment. It is described in the 1st paragraph of results section, total 1342 COVID-19 patients were identified who had completed 30 days after being diagnosed as COVID-19 positive by RT-PCR the during data collection period. The characteristics of the individuals excluded from the study were similar from those of included as the patients were from both sexes and all of them were aged >18 years. It was assessed by comparing their background characteristics obtained from the laboratory records. At the final stage, 1204 patients participated in the study, after excluding participants who had wrong contact number, didn`t attend the phone call, unwilling to participate and had an incomplete interview, from initially identified 1342 COVID-19 patients.

13. Comment: The authors have described the advantages of using CDC HRQOL-14 (lines 189-201). However, there are other generic HRQOL measures which apart from providing health profiles of the patients, also provides the index-values (e.g., in range of -1 to +1), hence facilitate the calculation of QALYs and conduct of cost-utility analyses. Thereby, the authors are requested to provide the rationale of using CDC HRQOL-14, over the other HRQOL instruments.

Response: Thank you for the vibrant comment. The rationale of using CDC HRQOL-14, over the other HRQOL instruments is given in the lines 113-123 of the ‘Introduction’ section and in the lines 519-539 of the ‘Discussion’ section.

14. Comment: Please discuss the validity of telephone administration of CDC HRQOL-14 by providing some relevant references.

Response: Thank you for the pertinent comment. Reference (Reference 13) for the validity of telephone administration of CDC HRQOL-14 is provided.

15. Comment: Please mention if there was any respondent who answered ‘no’ while responding to question 1 of the activity limitation module? May provide this information in Table 2.

Response: Thank you for the valuable comment. Question 1 of the activity limitation module (Duration of health related limited activity) was measured in days, which is a quantitative data. So, the respondents who answered ‘no’ to question 1 of the activity limitation module, their duration was recorded as ‘0’ days and accordingly, the mean value was calculated and provided in the Table 2.

16. Comment: Please comment in the results section if representativeness of the study population in comparison to the general population in Bangladesh was achieved. Which tests were used to assess the representativeness? The percentages of the larger population may be added to Table 1 to see the differences.

Response: Thank you for the vial comment. The finding elated to the representativeness of the study population in comparison to the general population in Bangladesh is added in the lines 288-292 of the results section. The percentages of the larger population is added in texts of the Table-1.

17. Comment: Provide the information regarding overall unhealthy days. May add this in Table-2.

Response: Thank you for the vital comment. Information regarding overall unhealthy days are added in Table-2.

18. Comment: Describe why the question 2 of the activity limitation module had been excluded?

Response: Thank you for the crucial comment. Reasons for exclusion of question 2 of the activity limitation module are described in the lines 212-215 under ‘Psychometric properties of the questionnaire’ of the ‘Materials and methods’ section.

19. Comment: Please comment on the overlap of physically unhealthy days and mentally unhealthy days. Report the number and % of the respondents who had reported only physically unhealthy days, only mentally unhealthy days, and both (along with extent of overlap).

Response: Thank you for the valuable comment. Table-2 is edited following the suggestions and related texts are rephrased accordingly in the lines 321-323 of the ‘Results’ section.

20. Comment: Provide details on the difference in the HRQOL among recovered and non-recovered COVID-19 patients (as mentioned in line118).

Response: Thank you for the crucial comment. Difference between recovered and non-recovered COVID-19 patients is given in the lines 134-135 in ‘Material and methods’ section.

21. Comment: Please discuss why the mean duration of healthy days was higher among the patients having a symptom and comorbidities as compared to their counterparts.

Response: Thank you for the very relevant comment. Actually, the mean duration of healthy days was lower among the patients having a symptom and comorbidities. So, information is corrected according to the results of the study in the lines 354–356 of the ‘Results’ section and discussed in the lines 487-489 of the ‘Discussion’ section.

22. Comment: A thorough copy-edit of the manuscript is required in terms of uniform use of abbreviations (using full-forms at the time of first use, and abbreviations at subsequent use), spelling and grammatical errors (e.g., lines 55, 56, 60, 94, 177, 232, 304-306, etc.), and table-footnotes (e.g., AOR in Table-3, use of single asterisk in Table-4).

Response: Thank you for the very important comment. Copy-editing of the manuscript is done properly considering your valuable suggestions.

23. Comment: In figure-2, anosmia may be placed before shortness of breath. Likewise, in figure-3, chronic liver disease may be placed before chronic kidney disease.

Response: Thank you for the specific comments. Based on the suggestions, figure 2 and figure 3 are edited.

Responses to comments of the reviewer 2 

1. Comment: The rationale of the study needs to be well elaborated.

Response: Thank you for the vital comment, Rationale of the study is now well elaborated.

2. Comment: Addition of studies using the same tool and specifically the used one will add more value.

Response: Study on measuring the HRQOL of COVID-19 patients using the CDC HRQOL-14 questionnaire is very scarce. So, we are could not add more relevant studies using the same tool for comparison.

3. Comment: The methodology needs to be technically sounder. The sample size calculation reference needs to be incorporated. How eligibility criteria were developed?

Response: Thank you for the vibrant comment. The ‘Methodology’ section is rewritten. Sorry, we could not add the reference for calculating sample size because sample size was based on the availability of all diagnosed COVID-19 patients on a selected month in a specific laboratory. The samples were selected purposively based on the specific selection criteria. For your kind attention, the selection criteria were developed based on the requirements of CDC HRQOL-14 questionnaire.

4. Comment: Results are quite confusing and need to be presented in a more understandable manner. The linear regression need to discussed in more detail with respect to different variables.

Response: Thank you for the pertinent comment. The ‘Results’ section is rewritten in an understandable manner.

5. Comment: Discussion need to be rewritten.

Response: Thank you for the vital comment. ‘Discussion’ section is rewritten and reorganized in a standard format.

6. Comment: Conclusion need to be rewritten with respect to study objectives.

Response: Thank you for the crucial comment. Conclusion is rewritten with respect to the study objectives.

7. Comment: The quality of writing in terms of English language is poor. The article needs major editing.

Response: Thank you for vital the comment. Comprehensive language editing of the article is done.

---

## [Decision Letter · Decision Letter 1]

22 Nov 2022

PONE-D-22-12676R1Health-related quality of life of the adult COVID-19 patients following one-month illness experience since diagnosis: findings of a cross-sectional study in BangladeshPLOS ONE

Dear Md. Ziaul Islam

Thank you for submitting your manuscript to PLOS ONE. After careful consideration, we feel that it has merit but does not fully meet PLOS ONE’s publication criteria as it currently stands. Therefore, we invite you to submit a revised version of the manuscript that addresses the points raised during the review process.

Apologies in the delay in providing you with an Editorial decision. 

To ensure the Editor and Reviewers will be able to recommend that your revised manuscript is accepted, please pay careful attention to each of the comments that have been pasted underneath this email. This way we can avoid future rounds of clarifications and revisions, moving swiftly to a decision. Please ensure that your decision is justified on PLOS ONE’s publication criteria and not, for example, on novelty or perceived impact.

We look forward to receiving your revised manuscript.

Kind regards,

Mohammad Hayatun Nabi, MBBS, MHSM, MPH, PHD

Academic Editor

PLOS ONE

Reviewers' comments:

Reviewer's Responses to Questions

**Comments to the Author**

1. If the authors have adequately addressed your comments raised in a previous round of review and you feel that this manuscript is now acceptable for publication, you may indicate that here to bypass the “Comments to the Author” section, enter your conflict of interest statement in the “Confidential to Editor” section, and submit your "Accept" recommendation.

Reviewer #1: All comments have been addressed

Reviewer #2: (No Response)

2. Is the manuscript technically sound, and do the data support the conclusions?

Reviewer #1: Yes

Reviewer #2: Partly

3. Has the statistical analysis been performed appropriately and rigorously? 

Reviewer #1: Yes

Reviewer #2: No

4. Have the authors made all data underlying the findings in their manuscript fully available?

Reviewer #1: Yes

Reviewer #2: Yes

5. Is the manuscript presented in an intelligible fashion and written in standard English?

Reviewer #1: Yes

Reviewer #2: No

6. Review Comments to the Author

Reviewer #1: (No Response)

Reviewer #2: I would like to thank the authors for revising the manuscript. Upon assessing the revised version, I request that the following issues must be addressed before considering it suitable for publication:

Comment 1: While responding to my Comment-5 on the previous version (How did the authors identify and manage the heterogeneity in the COVID-19 severity among the sample?), the authors have mentioned that they have included this as a limitation in the lines 206-209 under the heading of ‘Data collection’ of the ‘Methods’ section and in the lines 514-517 of the ‘Discussion’ section. However, the lines at these places does not convey this meaning. I would like to request again that it should be mentioned in the manuscript in the explicit language, so that readers can draw the interpretation accordingly.

Comment 2: While responding to my Comment-3 on the earlier version (how it was determined that a period of 30-days is appropriate for a COVID patient to get its HRQOL stabilised/ returned to baseline), authors have described the appropriateness of the questionnaire to capture the HRQoL in the last 30 days. However, my concern still remains unaddressed. I would highly appreciate if either it can be addressed, or explicitly mentioned as a limitation in the manuscript, so that the readers can interpret the results in a transparent manner.

Comment 3: It will be very beneficial to the readers to know that how the authors investigate the variability among the responses obtained by different interviewers? The results of this analysis may be reported in a table.

Comment 4: Please explain the rationale of citing FAQs on CDC HRQOL-14 (Reference no. 13) while describing the similarities between the sample and reference population (in the lines 288-290)?

Comment 5: In my Comment-12 on the earlier version, I asked if the (clinical) characteristics of the individuals excluded from the study were similar / different from those of included? This information is usually reported to explain that the analysis has not omitted the patients of a particular severity, and the results can be interpreted accordingly.

Comment 6: While responding to my Comment-20 on the earlier version (Provide details on the difference in the HRQOL among recovered and non-recovered COVID-19 patients), the authors have mentioned that “Difference between recovered and non-recovered COVID-19 patients is given in the lines 134-135 in ‘Material and methods’ section.” However, there is no mention of difference in HRQoL between recovered and non-recovered at this place. Secondly, please mention these differences in the ‘Results’ section, not in the ‘Methods’ section.

Comment 7: Please explain why the paper of Moriarty et al (The Center for Disease Control and Prevention's Healthy Days Measures–Population tracking of perceived physical and mental health over time) has been cited as different reference numbers at different places of the manuscript (first as 12, and then as 17)?

7. PLOS authors have the option to publish the peer review history of their article (what does this mean?). If published, this will include your full peer review and any attached files.

Reviewer #1: **Yes: **madeeha malik

Reviewer #2: **Yes: **Gaurav Jyani

---

## [Author Response · Author response to Decision Letter 1]

25 Nov 2022

Responses to comments of the reviewer 2

1. Comment: While responding to my Comment-5 on the previous version (How did the authors identify and manage the heterogeneity in the COVID-19 severity among the sample?), the authors have mentioned that they have included this as a limitation in the lines 206-209 under the heading of ‘Data collection’ of the ‘Methods’ section and in the lines 514-517 of the ‘Discussion’ section. However, the lines at these places does not convey this meaning. I would like to request again that it should be mentioned in the manuscript in the explicit language, so that readers can draw the interpretation accordingly.

Response: Thank you for the crucial comment. The reasons for not being able to identify and manage the heterogeneity in the COVID-19 severity among the sample is mentioned in the manuscript in explicit language in the lines 442-446 of the ‘Discussion’ section.

2. Comment: While responding to my Comment-3 on the earlier version (how it was determined that a period of 30-days is appropriate for a COVID patient to get its HRQOL stabilized / returned to baseline), authors have described the appropriateness of the questionnaire to capture the HRQoL in the last 30 days. However, my concern remains unaddressed. I would highly appreciate if either it can be addressed, or explicitly mentioned as a limitation in the manuscript, so that the readers can interpret the results in a transparent manner.

Response: Thank you for your valuable comment. For your kind information, due to the lack of any previous baseline data, we could not determine whether the HRQoL of a COVID-19 patient stabilized or returned to baseline after 30 days. We only assessed the HRQoL of the COVID-19 patients for the period of one month since diagnosis (whether it was stabilized / returned to baseline or not) following the specific guidelines of the questionnaire, CDC HRQOL-14. This is added as a limitation of the study in the lines 451-457 of the ‘Discussion’ section. In addition, we compared the HRQOL by the background (not baseline) characteristics of the patients. Accordingly, corrections are made by using the term ‘background’ instead of ‘baseline’ characteristics (Tables 3, 4 and 5). 

3. Comment: It will be very beneficial to the readers to know that how the authors investigated the variability among the responses obtained by different interviewers? The results of this analysis may be reported in a table.

Response: Thank you for the vibrant and valuable comment. For your kind information, we trained the interviewers extensively on telephone interviews and also conducted shadow interviews to minimize variability among the responses obtained by different interviewers which is mentioned in the lines 159-161 of the ‘Data Collection’ under the methods section.

4. Comment: Please explain the rationale of citing FAQs on CDC HRQOL-14 (Ref. no. 13) while describing the similarities between the sample and reference population (Lines 288-290)?

Response: Thank you for the crucial comment. For your kind consideration, that reference (Ref. no. 14) was cited mistakenly. The reference is corrected (Page 6) and no explanation for the rationale of citing FAQs is not required. Other references are also re-checked and accordingly those are rearranged throughout the article and in the reference list.

5. Comment: In my Comment-12 on the earlier version, I asked if the (clinical) characteristics of the individuals excluded from the study were similar / different from those of included? This information is usually reported to explain that the analysis has not omitted the patients of a particular severity, and the results can be interpreted accordingly.

Response: Thank you for your crucial comment. We collected data through telephone interviews and there was no scope to investigate and compare the clinical characteristics of the individuals excluded from the study were similar / different with those who were included. As a result, we it was not possible to be concerned about omitting the patients with particular severity. For example, some individuals were excluded who didn’t response to our telephone calls for three separate occasions and whose telephones were inactive. So, it was not possible to collect data on the severity form those excluded individuals and to compare it with the individuals who were included. It is mentioned in the lines 130-136 under the heading of study population in the ‘Materials and methods’ section.

6. Comment: While responding to my Comment-20 on the earlier version (Provide details on the difference in the HRQOL among recovered and non-recovered COVID-19 patients), the authors have mentioned that “Difference between recovered and non-recovered COVID-19 patients is given in the lines 134-135 in ‘Material and methods’ section. However, there is no mention of difference in HRQoL between recovered and non-recovered at this place. Secondly, please mention these differences in the ‘Results’ section, not in the ‘Methods’ section.

Response: Thank you for the valuable comment. For your kind information, we collected data on HRQOL from the COVID-19 patients who completed 30 days since diagnosis whether they were recovered or not-recovered. We had no data related to recovery because it was not possible to collect those data through telephone interviews without clinical evidences. So, we didn’t examine and compare the difference in the HRQOL between recovered and non-recovered COVID-19 patients separately. Moreover, this was not a primary concern of our study, hence, we didn’t conduct any analysis to find out the difference in the HRQO and mention in the results. 

7. Comment: Please explain why the paper of Moriarty et al (The Center for Disease Control and Prevention's Healthy Days Measures–Population tracking of perceived physical and mental health over time) has been cited as different reference numbers at different places of the manuscript (first as 12, and then as 17)?

Response: Thank you for the dynamic comment. We are extremely sorry for this error. The reference no. 17 was a repetition of ref. no. 12. So , the ref. No 17 is deleted followed by, all the references are re-checked and rearranged throughout the article and in the reference list (Pages 24-25).

---

## [Editor Report · Decision Letter 2]

29 Nov 2022

Health-related quality of life of the adult COVID-19 patients following one-month illness experience since diagnosis: findings of a cross-sectional study in Bangladesh

PONE-D-22-12676R2

Dear Dr. Md. Ziaul Islam,

We’re pleased to inform you that your manuscript has been judged scientifically suitable for publication and will be formally accepted for publication once it meets all outstanding technical requirements.

Kind regards,

Mohammad Hayatun Nabi, MBBS, MHSM, MPH, PHD

Academic Editor

PLOS ONE
---

## [Editor Report · Acceptance letter]

10 Feb 2023

PONE-D-22-12676R2 

Health-related quality of life of the adult COVID-19 patients following one-month illness experience since diagnosis: findings of a cross-sectional study in Bangladesh 

Dear Dr. Islam:

I'm pleased to inform you that your manuscript has been deemed suitable for publication in PLOS ONE. Congratulations! Your manuscript is now with our production department. 

Kind regards, 

on behalf of

Dr. Mohammad Hayatun Nabi 

Academic Editor

PLOS ONE